# Carboxylated Graphene Oxide (c-GO) Embedded ThermoPlastic Polyurethane (TPU) Mixed Matrix Membrane with Improved Physicochemical Characteristics

**DOI:** 10.3390/membranes13020144

**Published:** 2023-01-22

**Authors:** Muhammad Zahid, Maryam Saeeda, Nimra Nadeem, Hafiz Muhammad Fayzan Shakir, Waleed A. El-Saoud, Osama A. Attala, Kamal A. Attia, Zulfiqar Ahmad Rehan

**Affiliations:** 1Department of Chemistry, University of Agriculture, Faisalabad 38040, Pakistan; 2Department of Textile Engineering, School of Engineering and Technology, National Textile University, Faisalabad 37610, Pakistan; 3School of Materials Science and Engineering, Northwestern Polytechnical University, Xi’an 710060, China; 4Natural Hazards Research Unit, Department of Environmental and Health Research, Umm Al-Qura University, Mecca 21955, Saudi Arabia; 5Department of Environmental and Health Research, The Custodian of the Holy Mosques Institute for Hajj and Umrah Research, Umm Al-Qura University, Mecca 21955, Saudi Arabia; 6Biology Department, Al-Jammoum University College, Umm Al-Qura University, Makkah 24381, Saudi Arabia; 7Department of Materials, National Textile University, Faisalabad 37610, Pakistan

**Keywords:** thermoplastic polyurethane (TPU), carboxylated graphene oxide (c-GO), nanocomposite membrane, water flux, dye rejection

## Abstract

Water is an important component of our life. However, the unavailability of fresh water and its contamination are emerging problems. The textile industries are the major suppliers of contamination of water, producing high concentrations of heavy metals and hazardous dyes posing serious health hazards. Several technologies for water purification are available in the market. Among them, the membrane technology is a highly advantageous and facile strategy to remediate wastewater. Herein, the distinguished combination of pore-forming agents, solvent, and nanoparticles has been used to achieve improved functioning of the polymeric composite membranes. To do so, graphene oxide (GO) was fabricated via Hummer’s technique and GO functionalization using chloroacetic acid (c-GO) was performed. Thermoplastic polyurathane (TPU) membranes having different concentrations c-GO were made using the phase inversion technique. Scanning electron microscopy (SEM), Fourier transforms infrared spectroscopy (FT-IR), and X-ray diffraction (XRD) was used to examine surface morphology, chemical functionalities on membranes surfaces, and crystallinity of membranes, respectively. The temperature-dependent behavior of c-GO composite membranes has been analyzed using DSC technique. The water contact angle measurements were performed for the estimation of hydrophilicity of the c-GO based TPU membrane. The improved water permeability of the composite membrane was observed with increasing the c-GO concentration in polymeric membranes. c-GO was observed as a potential candidate that enhanced membrane physicochemical properties. The proposed membranes can behave as efficient candidates in multiple domains of environmental remediation. Furthermore, the improved dye rejection characteristics of proposed composite membranes suggest that the membranes can be best suited for wastewater treatment as well.

## 1. Introduction

Pure water is essential and a major component for human survival on this planet [1]. Drinking water, both in terms of quality and quantity, has become a critical challenge for human survival [2]. The expansion of industries, agricultural resources, and the population growth all demands access to clear water [3]. The textile industries are the major contributor of colored wastewater production [4,5]. It is the biggest zone that is spreading extreme water pollution. Chemical additives and various types of dyes are also present in textile wastewater. The problem of colors in textile effluent and the difficulties related to the degradation of dye wastewater is alarming [6,7,8,9,10]. There is a need to design solutions that are cost effective and time efficient to recover and provide high quality of water.

Membrane technology is used in a number of water treatment processes for this purpose, including municipal and industrial wastewater recycling, and also sea- and brackish water desalination [11,12]. A membrane is a thin sheet or layer that serves as a selective barrier between two phases (liquid, vapor, or gas), allowing certain components to flow through depending on its structure [13,14]. Membranes are advantageous because of their low production cost, energy efficiency, ease of fabrication, and lack of thermal treatment [15]. Polymeric membranes have drawn a lot of interest due to their high-water flux, environmental friendliness, cost-effectiveness, and energy efficiency. Polymer-based membranes are composed up of polymers and organic matter including polyether sulfone, polysulfone, polyether ketone, thermoplastic polyurethane (TPU), cellulose acetate (CA), polyimide (PI), polysulfide (PS) (Espinoza et al., 2014) for treatment of wastewater. Most polymeric membranes are either hydrophobic or have low hydrophilicity. TPU (thermoplastic polyurethane) is an engineered thermoplastic polymer with good mechanical characteristics. TPU network structure is made up of soft and rigid segments, giving it qualities resembling rubbery and unstructured thermoplastics. To improve the features of TPU such as hydrophilic nature, smooth surface, the charge on the surface, and antibacterial activity, selective advancement in the membrane could be carried out to increase water permeability (Riaz et al., 2016). Functionalized nanoparticles and pore-forming agents are gaining attention of scientific community to improve the physicochemical characteristics of TPU. Due to its excellent mechanical characteristics, high surface area, ease of creation of dense membrane structures, and superior water dispersal ability graphene oxide (GO) have emerged as a potential membrane nanomaterial [16]. Multiple oxygen functional groups in GO such as COOH, epoxy, carbonyl, and -OH, improve its hydrophilicity and provide remarkable characteristics [17,18,19,20,21,22,23]. Functionalized carboxylated graphene oxide-based membranes may be used to remove ionic dyes [24], and changing hydroxyl groups and ether linkages into carboxyl groups might increase the membrane’s performance.

Here we proposed the fabrication of novel carboxylated graphene oxide polyurethane membrane (c-GO/TPU) by NIPS method. Different concentrations of c-GO were selected to form TPU based composite membranes. All the membranes were well characterized in term of surface morphology using SEM, surface functional groups using FTIR, and membranes crystallinity using XRD analysis. Moreover, the mechanical strength of pristine TPU and c-GO embedded TPU membranes were checked using tensile strength estimation approach. The pure water flux potential and dye rejection efficiency of membranes were also analyzed. The effect of c-GO content addition on the hydrophilicity of membranes was analyzed using water contact angle measurements. The DSC approach was accessed to analyze the thermal characteristics of selected membranes. The extensive analysis of c-GO based TPU composite membranes suggest that the addition of c-GO content improves the surface morphology, hydrophilicity, and mechanical strength characteristics. Therefore, depending upon the application area, the proposed membranes can be an efficient candidate in multiple domains of environmental remediation. Furthermore, the improved dye rejection characteristics of proposed composite membranes suggest that the membranes can be best suited for wastewater treatment as well.

## 2. Materials and Methods

### 2.1. Materials

Thermoplastic polyurethane was obtained from Townsend Corporation (RE-FLEX585-XU, MW = 25KD). Graphite powder (average particle size of <20 µm), and phosphoric acid (H_3_PO_4_) were bought from Sigma-Aldrich. Potassium permanganate (KMnO_4_ Mw ~158.03, Assay = 99.5%) was purchased from AnalaR^®^. Sodium hydroxide pellets (NaOH: 98% Analytic-ACS) were purchased from ICON CHEMICAL. Polyvinyl pyrrolidone (PVP of Mw ~40,000 Da), sulfuric acid (H_2_SO_4_ Assay 98.5%), Hydrogen peroxide (H_2_O_2_; 30%) aqueous solution, and N, N-Dimethyl formamide (DMF) were purchased from DAEJUNG. Chloroacetic acid (C₂H₃C.lO₂: 94.49 g/mol^−1^) was purchased from Merck, Germany. Deionized water (DI) used in this study was produced through an integrated system of a Milli-Q (Merck Millipore, Ireland). All compounds were of analytical rank and were used exactly as directed.

### 2.2. Synthesis

#### 2.2.1. Preparation of Graphene Oxide

Graphene oxide was fabricated using modified Hummer’s method. The detailed process is described somewhere else [25].

#### 2.2.2. Synthesis of Carboxylated Graphene Oxide (c-GO)

Carboxylated graphene oxide was synthesis by the reaction of chloroacetic acid with sodium hydroxide. For this 0.5 g of graphene oxide was added into 250 mL distilled water and sonicated for 1 h to form suspension. After complete dispersion, 3.6 g of sodium hydroxide was added, and solution was again sonicated for 1 h. Then, the 10 g of chloroacetic acid was added into the reaction mixture and ultra-sonicated for 1 h. The obtained c-GO was washed several times with DI water until the pH of the solution becomes neutral. The c-GO was then dried in the oven overnight at 70 °C.

#### 2.2.3. Fabrication of Carboxylated Graphene Oxide Based TPU Membranes

Carboxylated graphene oxide TPU based membranes were fabricated by phase inversion method. In the first step, c-GO was sonicated in DMF for 1 h 15 percent TPU was dissolved in DMF to make the membrane casting solution. Under magnetic stirring, different quantities of c-GO (Table 1.) with fixed number of pore-forming agents (PVP) was introduced to the TPU solution for full dissolution. The solution was then allowed to stir for 4 to 6 h to remove air bubbles and to make it homogeneous. The obtained solution was cast on a 500 µm thick film applicator. The casted membrane was immersed into the distilled water bath for solvent exchange. After two minutes, the c-GO embedded TPU membrane was detached from the casting plate. The membrane was dried at room temperature. All the membranes with different concentrations of carboxylated graphene were fabricated using similar approach. The schematic representation of membrane casting process is presented in Figure 1.

### 2.3. Characterization of GO and c-GO and TPU Composite Membranes

Characterization of fabricated composite membrane was carried out by different techniques such as FE-SEM, XRD, FTIR spectroscopy, contact angle and tensile strength. Field emission scanning electron microscopy was used to examine the surface morphology of membranes (FEI, Quanta FEG 450). Membrane samples were copper taped to the grid and sputtered with gold using a sputter coater (Quorum Q150R ES, Quorum Technologies Ltd., Ashford, Kent, England). The presence of functional groups at different blend and solution formulations was identified using FTIR spectroscopy (FTIR; Bruker Tensor 27, Billerica, MA, USA). XRD analysis was used to examine the micro sheet samples (Bruker-AXS diffractometer, Karlsruhe, Germany). A goniometer OCA15EC was used to measure the contact angle of water on membrane surfaces using the sessile drop method at ambient temperature (Dataphysics, Karlsruhe, Germany). Thermal analysis was conducted at a 10 °C/min heating rate under nitrogen atmosphere using DSC technique.

### 2.4. Measurement of Water Contact Angle, Pure Water Flux and Dye Rejection Rate

The contact angle was measured using a contact angle measuring device and the sessile-drop technique. A droplet of water was placed on the membrane surface and the contact angle of the droplet with the surface was computed using this approach. The contact angle was measured at five random points on each membrane to minimize the research error, and the average of results is reported. A cross-filtration equipment was used to assess the membrane’s pure water flow.

Hydrophilic properties of all fabricated membranes were examined by Attension Theta Tensiometer using measurement of water contact angle. To do so, water was utilized by means of probe liquid. The fixed contact angle (made between the drop of water and the surface of membrane) was determined at room temperature via the sessile drop mode using goniometer device. The contact angle was calculated by averaging five measurements performed at various points across the surface of membrane [11,18].

The tensile strength of the c-GO based membranes were measure using Instron tensile test equipment. Membranes were cut into a standard form prior to testing. The thermal characteristics of c-GO/TPU membranes were determined using differential scanning calorimeter (DSC 250 from TA, New Castle, USA) instrument. Prepared membranes weighing 10 mg were encapsulated in an aluminum pan and tested at temperatures ranging from 25 to 300 degrees Celsius at a rate of 10 degrees Celsius per minute under a N_2_ environment [11].

A lab-scale cartridge filter was used at cross-end mode operation to examine the permeate flux of water as well as rejection of dye by c-GO/TPU membranes. Pressure gauges are the major constituents of the filtration system, a 2-L feed tank (having mixer and temperature control), low as well as high pressure feed pumps (1–15 bar), and stainless-steel flat membrane module (effective area of 8.6 cm^2^) were equipped with the reaction system. Before measurements of flux, membranes remained soaked in distilled water for 24 h. The membranes were then compacted for 30 min at less than 2 bar of distilled water until a steady flow was attained. The pressure instantly decreased to 1 bar, and a pure water flux test was carried out for 1 h. The volume of filtrate was collected and measured after every 5 min. In conclusion, the Equation (1) was used to compute the flow [26].
(1)$$J=\frac{V}{A\Delta T}$$
where *V* stands for the permeated volume of pure water (L), *A* is the operative membrane area (m^2^), Δ*T* stands for the sampling time (h) and *J_w_* is the pure water flux (L/m^2^ h).

The dye rejection potential of all membranes was analyzed using aqueous solution of Coomassie Blue dye. The feed solution was prepared by 0.01 g/cm^3^ Coomassie Blue in ethanol. This solution was used as feed solution to make 2 L of aqueous solution with the dye concentration of 10 ppm at neutral pH (25 °C). The formula given Equation (2) below was used to calculate dye rejection [27].
(2)$$R=\left(1-\frac{C_{p}}{C_{f}}\right)\times 100$$

In above equation *R* is membrane rejection, *C_f_* is dye concentration in the feed and *C_p_* is dye concentration at permeate side.

### 2.5. Membrane Porosity Determination Using Gravimetric Method

Gravimetric method was used to determine the membrane porosity. To do so, all membranes were oven dried and weighed before experiment. The membranes were then immersed in kerosene oil for 24 h and reweighed. The average porosity was determined in terms of overall void fraction which can be calculated as the pore volume divided by the total membrane volume. The average porosity was calculated using the formula given in Equation (3):(3)$$\varepsilon_{m}\;\%=\left[\frac{W_{1}-W_{2}}{D_{k}}\right]/\left[\frac{W_{1}-W_{2}}{D_{k}}+\frac{W_{2}}{D_{pol}}\right]\times 100$$
where
$$\begin{array}{c} W_{1}=weight\;of\;wet\;membrnae,W_{2}=weight\;of\;dry\;membrane,\\ D_{k}=denisty\;of\;kerosene\;oil\;\left(0.82\frac{g}{{\mathrm{cm}}^{3}}\right),\;D_{pol}=density\;of\;polymer \end{array}$$

## 3. Result and Discussion

### 3.1. Characterization of Nanoparticles

The FTIR spectral information is presented in Figure 2. Figure 2 presents the FTIR spectra of GO and c-GO. The stretching vibrations of the hydroxyl (O-H) groups upon graphene oxide are represented by a broad peak in the upper-frequency band of 3445 cm^−1^ [28,29]. The absorption peaks at 2850 cm^−1^ show the stretching vibration of CH_2_. The ketone group is responsible for the appearance of a peak at 1627 cm^−1^, and sp2 hybridization is responsible for the primary graphitic domain of the peak at 1544 cm^−1^. The C-O is shown by the band at 1457 cm^−1^, whereas the C-O stretching of epoxy groups is indicated by the band at 1243 cm^−1^. The C-O-C stretching of alkoxy groups is revealed by the mode at 1087 cm^−1^. Compared to spectra of GO and c-GO, the vibration band of C-O-C has observed on the 1052 cm^−1^ and OH shows absorption peak on the 1349 cm^−1^. The absorption peak of C=O was around 1620 cm^−1^ become broader and high strength. This peak was overlapped the –COOH peak at 1720 cm^−1^. Between 3000 and 3500 cm^−1^, a hydroxyl group (-OH) adsorption band was identified, indicating a substantial quantity of carboxyl group on the graphene surface. the results also support clear evidence about successful formation of c-GO as the peaks corresponding to carboxylic groups i.e., peaks at 1728 cm^−1^ and 1614 cm^−1^ showed clear enhancement in C-GO. Similarly, the peak ascribed to aromatic group also showed clear enhancement in c-GO [30].

Figure 2b represents the FTIR spectra of thermoplastic polyurethane membrane, PVP-based TPU membranes, and TPU membrane with different concentrations of carboxylated graphene oxide. The FTIR spectra of TPU exhibit a transmittance peak at 3330 cm^−1^ corresponds to the –NH gorup. The peak positioned at 2960 cm^−1^ is ascribed to the CH_2_. Another peak at 1215 cm^−1^ is representative of the C-N-H bonding. The functional groups of C-O-C in TPU exhibits peak at 1120 cm^−1^. The spectrum of TPU-PVP shows absorption bands at 1639 cm^−1^ and 1215 cm^−1^ which denote carbonyl and amide groups. Carboxylated graphene oxide-based TPU membranes exhibit peaks at 3330 cm^−1^, 2960 cm^−1^, 1687 cm^−1^, 1516 cm^−1^, 1215 cm^−1^, and 1120 cm^−1^. All these peaks are present in the carboxylated graphene oxide ensuring succesful insertion of c-GO into thermoplastic polyurethane membranes matrix.

#### 3.1.1. SEM Analysis

SEM investigation confirms the surface morphology of TPU and TPU composite membranes, as shown in Figure 3. The SEM pictures of TPU composite membranes with varying quantities of c-GO content demonstrate the presence of particles on the membrane’s surface. Water addition during casting solution preparation may have generated a rough picture that correlates to fractures and ridges in c-GO 4 (0.5 wt percent). Particle aggregates appear on the surface of the c-GO 4 membrane at intervals, indicating particle dispersion inside the polymeric substance. The insertion of c-GO in pristine TPU (Figure 3a) results extraordinary improvement in morphology of composite membranes. The improved morphology is attributed to the increased surface roughness owing to the nano-dimensional c-GO loading. High surface roughness is effective is generating surface defects which contribute towards improved adsorption properties of membranes as well.

#### 3.1.2. XRD

Figure 4 presents the XRD analysis of all c-GO based TPU membranes. A broad peak in the range of 11°–20° is ascribed to the amorphous nature of TPU. Insertion of c-GO into the matrix of TPU dose not significantly changed the peak structure. However, peak broadening supports the incorporation of c-GO. C-GO exhibits characteristic peak around 2θ = 10° [31], no clear indication of peaks located at 2θ = 10° in composite membrane suggests that c-GO was well incorporated with TPU matrix [32].

#### 3.1.3. Contact Angle

Membrane contact angle measurements are commonly used to determine membrane surface hydrophilicity. Figure 5 presents that the contact angle of pristine TPU membrane is 76.95 decreased with the addition of different concentrations of carboxylated graphene oxide into membranes to 68.86, 61.01, 57.97, and 54.00 for c-GO 1, c-GO 2, c-GO 3, and c-GO 4, respectively. This suggested that the surface hydrophilicity has improved as a result of the addition of c-GO micro sheets, as previously observed [33] and corroborated by the pure water flow. A huge number of the micro sheets’ –COOH groups orient themselves towards the water, inducing hydrophilic characteristics on the membrane, which promotes water adsorption and hence improves water permeability [34].

#### 3.1.4. DSC

DSC curves of prepared TPU/c-GO membranes with different concentrations in this study are given in Figure 6. The Tg (Glass transition temperature) is generally used for interpretation of membrane structure while employing the thermal analysis upon membrane. The glass transition temperature of the pristine TPU is greater than the TPU/c-GO composite membranes (i.e., c-GO 0 and c-GO 4). With the addition of the carboxylated graphene oxide Tg of TPU decreased slightly i.e., 204 °C for pristine TPU to 202 °C and 201 °C for c-GO 0 and c-GO 4, respectively. Aside from c-GO, the reduction may be attributed to PVP’s plasticization effect since PVP can be used as a stabilizer and plasticizer for polymers [35]. The slight thermal degradation of c-GO 0 and c-GO 4 could be due to the interactions of c-GO and PVP with TPU network and good thermal conductivity of c-GO resulting in creation of degradation centers in composite membranes.

#### 3.1.5. Pure Water Flux of Membranes

One of the most important characteristics to consider when evaluating the performance of a manufactured membrane is its permeability to pure water. Figure 7 showed that with the addition of c-GO to the membrane matrix, the penetration fluxes of the membranes were improved. The water flux of the pure TPU membrane was 21.81 LMH and c-GO 4 was 142 LMH. Due to the addition of c-GO to membrane, porosity increase that causes high water flux. The oxygen-rich functional groups of the c-GO that have migrated to the top surface of the membranes have also increased the membranes’ hydrophilicity. Furthermore, c-GO nanofillers demonstrated a reduction in contact angle, which might help to improve water permeability [36]. Furthermore, the results of membrane porosity support the conclusion that increased membrane porosity by increasing c-GO content helps improve water flux (Table 2).

#### 3.1.6. Dye Rejection by Composite Membrane

Due to its excellent removal performance, and low operating cost, many types of membrane technology have been used to remove synthetic colours from wastewater. The dye rejection rate of the membrane rises when nanoparticles are added to it (due to improved morphology). Therefore, the dye rejection experiment was performed and results are presented in Figure 7. Increasing the c-GO content in TPU matrix improved the surface morphology in term of surface defect thereby imporving the adsoption capability of composite membranes which results in high dye rejection [37].

The results of membranes porosity calculated by gravimetric analysis is presented in Table 2. A rapid increase in membrane porosity was observed when pristine TPU membranes were composited with c-GO. The membrane porosity of pristine TPU was observed as 72.48% which increases to 87.54% with the addition of c-GO (1 wt %). Increased water flux can be correlated with the improved structure in term of membrane porosity. Among c-GO based TPU membranes an increasing trend of membrane porosity was observed by increasing c-GO content which also support the porosity induction is due to c-GO insertion.

## 4. Conclusions

In conclusion, the phase inversion method was used to produce TPU membranes containing variable c-GO content. All the membranes were well characterized in term of SEM, FTIR, and XRD analysis. Moreover, the mechanical strength of pristine TPU and c-GO embedded TPU membranes were checked using tensile strength estimation approach. The pure water flux potential and dye rejection efficiency of membranes were also analyzed. The effect of c-GO content addition on the hydrophilicity of membranes was analyzed using water contact angle measurements. With the addition of c-GO nanoparticles to the TPU polymer matrix, water absorption was increased while the contact angle was reduced. c-GO-4 had the highest water flow while having a lowest contact angle among all membranes. The DSC approach was accessed to analyze the thermal characteristics of selected membranes and little reduction in Tg was observed when TPU was composited with c-GO (owing to the degradation centers generation due to good thermal conductivity of c-GO). The extensive analysis of c-GO based TPU composite membranes suggest that the addition of c-GO content improves the surface morphology, hydrophilicity, and mechanical strength characteristics. Therefore, depending upon the application area, the proposed membranes can behave as efficient candidate in multiple domains of environmental remediation. Moreover, the improved dye rejection characteristics of proposed composite membranes suggest that the membranes can be best suited for wastewater treatment as well.

## Figures and Tables

**Figure 1 membranes-13-00144-f001:**
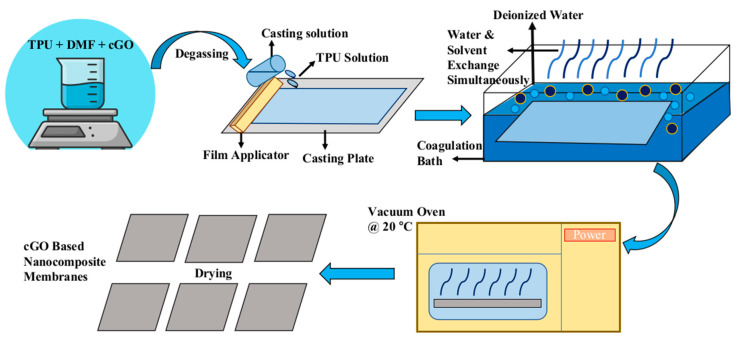
Schematic representation of phase inversion method.

**Figure 2 membranes-13-00144-f002:**
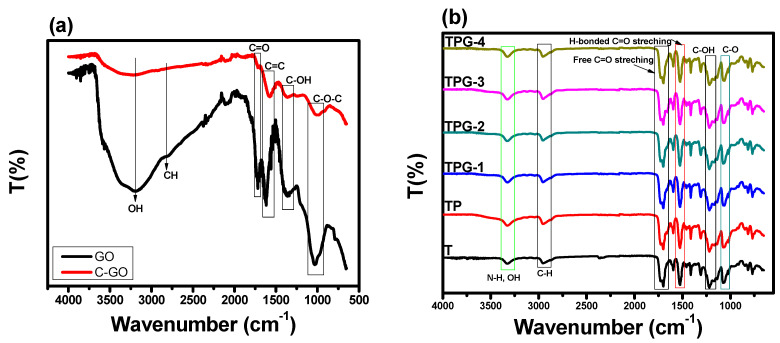
FTIR spectra of (**a**) GO and c-GO, (**b**) pristine TPU and c-GO embedded TPU membranes.

**Figure 3 membranes-13-00144-f003:**
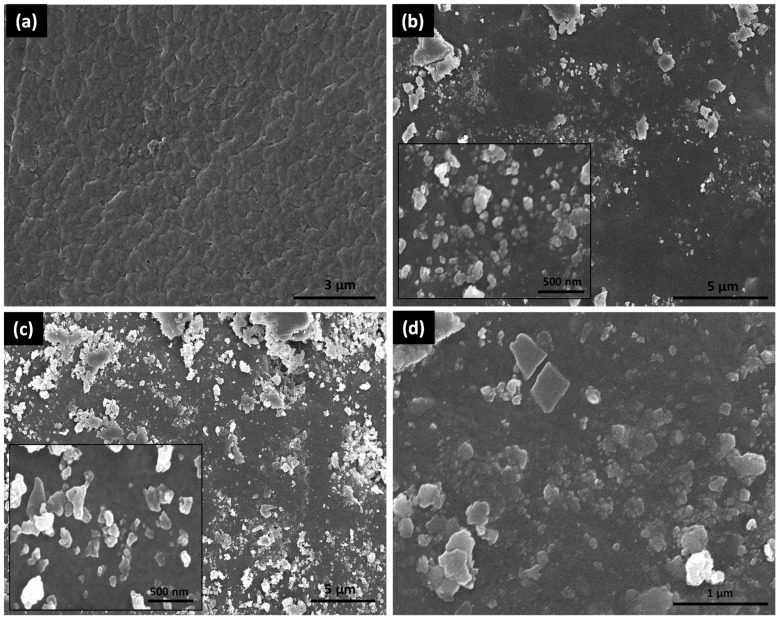
SEM images of (**a**) TPU (**b**) c-GO 1, (**c**) c-GO 3, and (**d**) c-GO 4.

**Figure 4 membranes-13-00144-f004:**
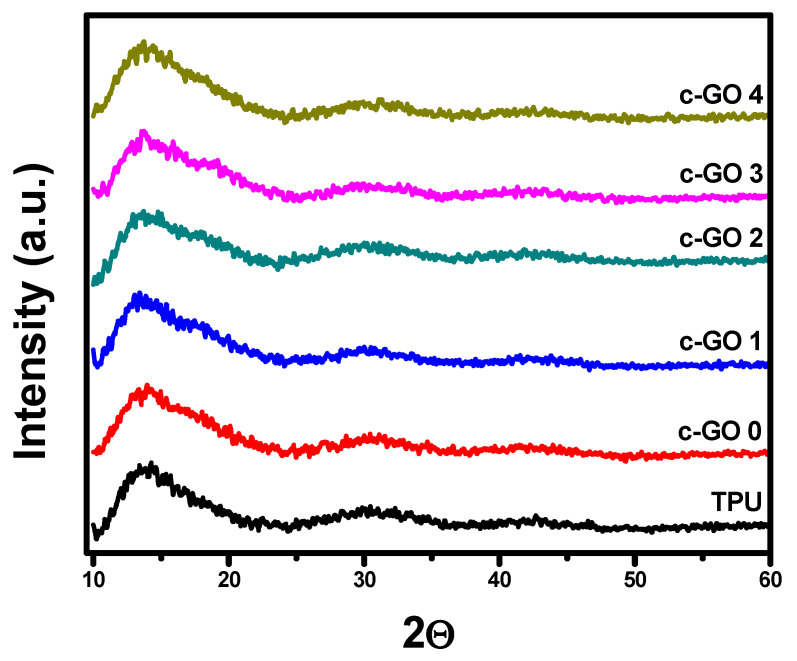
XRD pattern of TPU, and c-GO based TPU membranes.

**Figure 5 membranes-13-00144-f005:**
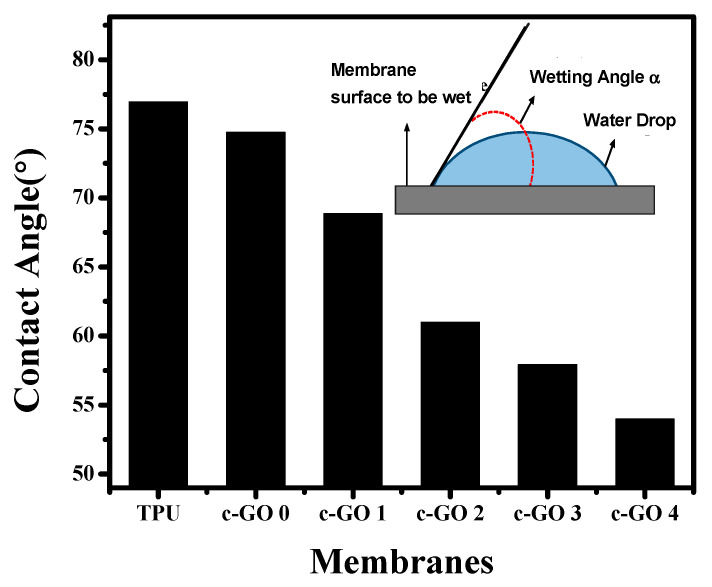
The contact angle of TPU and composite membranes.

**Figure 6 membranes-13-00144-f006:**
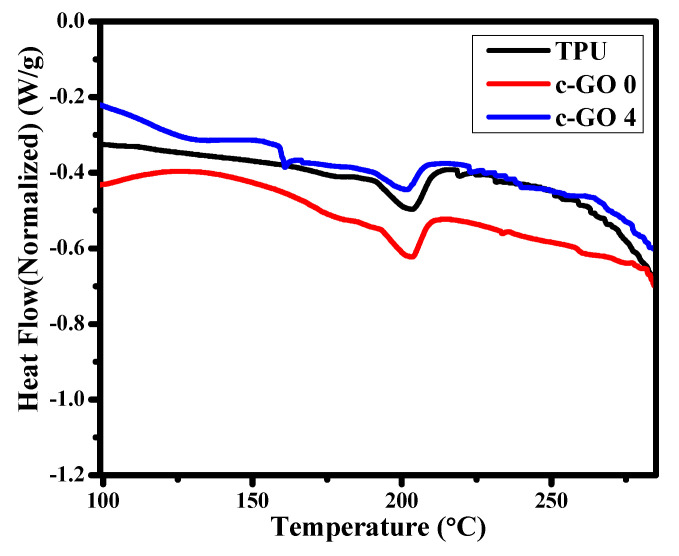
DSC thermogram of pristine TPU and TPU based composite membranes (c-GO 0 and c-GO4).

**Figure 7 membranes-13-00144-f007:**
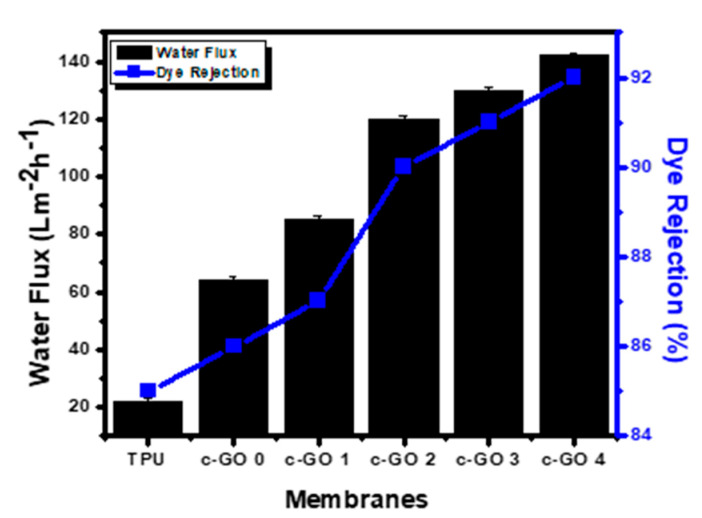
Pure water flux and dye rejection performance of TPU and c-GO/TPU membranes (Dye Coomassie Blue (10 ppm), pH = 7, T = 30 °C).

**Table 1 membranes-13-00144-t001:** Compositions of c-GO based TPU membranes.

Membrane	TPU (wt %)	PVP	DMF	c-GO
TPU	15%	0%	85%	-
c-GO 0	15%	0%	84.9%	0.1%
c-GO 1	15%	5%	79.8%	0.2%
c-GO 2	15%	5%	79.7%	0.3%
c-GO 3	15%	5%	79.6%	0.4%
c-GO 4	15%	5%	79.5%	0.5%

**Table 2 membranes-13-00144-t002:** Porosity (%) of TPU based c-GO composite membranes.

Sr#	Membranes	Porosity (%)
1	TPU	72.48
2	c-GO 0	74.70
3	c-GO 1	87.54
4	c-GO 2	88.53
5	c-GO 3	90.42
6	c-GO 4	91.26

## Data Availability

Not applicable.

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
