# Peer review of "Carboxylated Graphene Oxide (c-GO) Embedded ThermoPlastic Polyurethane (TPU) Mixed Matrix Membrane with Improved Physicochemical Characteristics"

_membranes, 2023, doi:10.3390/membranes13020144_

Round 1
Reviewer 1 Report
The manuscript “Carboxylated Graphene Oxide (c-GO) Embedded Thermo Plastic Polyurethane (TPU) Mixed Matrix Membrane with Improved Physicochemical Characteristics” describes the results of the fabrication of novel carboxylated graphene oxide polyurethane membrane (c-GO/TPU) by NIPS method. The manuscript sounds superficial with no important discussion of the obtained results and the discussion section requires the addition of some important information. Also, the experimental part did not show the most important information to make this work reproducible. Also, some parts require English proofreading. I have made several mandatory corrections to this work, which may improve its quality to be accepted for publication and support my decision (major revisions).
Line 35: Please change “The addition of c-GO composite,” to “By the addition of c-GO composite,”.
Lines 93-94: Please change “the proposed membranes can be have as efficient candidate ….” to “the proposed membranes can be an efficient candidate ….”.
Line 100: Please mention the molecular weight of the used polymer.
Line 109: What were the “Additional compounds ….” that the authors mentioned?
Section 2.2.2. requires extensive proofreading to be cohesive and understood. Please avoid too short sentences.
Section 2.2.3. Please use the abbreviation of N-methyl-2-pyrrolidone as mentioned in section 2.1.
Section 2.2.3.: Did the authors fabricate a membrane using 5% of PVP and 0.1% of c-GO?
Sections 2.3 and 2.4: Please use the abbreviations of the characterization techniques used as that mentioned in the abstract section.
Line 163-164: Please change “made among the drop of water and the surface of membrane” to “made between the drop of water and the surface of membrane”.
Section 2.4: Please mention the dye name, dye concentration, solution temperature and pH used to examine the membrane performance.
Lines 182-183: Please write equation 1 and refer to it within the manuscript. Give a reference to equation 1. The following reference can be used:
* Preparation and application of polyethersulfone ultrafiltration membrane incorporating NaX zeolite for lead ions removal from aqueous solutions. Desalin. Water Treat, 3, 28-56.
Section 2.4: Please write the equation used to calculate the dye rejection% and refer to it within the manuscript. Give a reference to this equation and the following reference can be used:
* Implementation of hierarchically porous zeolite-polymer membrane for Chromium ions removal. In IOP Conference Series: Earth and Environmental Science (Vol. 779, No. 1, p. 012099). IOP Publishing.
Lines 205-213: These lines require extensive proofreading to be cohesive and understood. Please avoid too short sentences.
Section 3.1.1. the results presented in this section require deep discussion. The scale bars are not readable. It is important to show the effect of adding cGO on the formation and shape of the formed pores, therefore images showing the cross-section of membranes are required.
Section 3.1.2. This section requires extensive proofreading to be cohesive and understood. Please use the same font size. Check the discussion with the presented results.
-In general, the discussion of the results did show the effect of different loading of cGO on the fabricated membranes and their properties.
Section 3.1.4. The authors discussed the effect of adding PVP on the Tg, however, they should discuss the effect of different loading of cGO on Tg of the fabricated membranes.
Section 3.1.5: This section requires extensive proofreading to be cohesive and understood.
Section 3.1.6: The results discussed in this section are different from those shown in figure 7. Please check them and their discussion.
Figure 7: The authors used different codes for the prepared membranes from those given in table 1. Please consistency is required. Also, mention the run conditions and dye name in the caption of the figure.
-Please correct the conclusion section according to the given comments. Also, please delete “energy sector” because the fabricated membranes were not tested in the energy sector; and apply this comment in the whole manuscript.
Reviewer 2 Report
In the manuscript, the authors intended to fabricate a novel membrane by introducing carboxylated graphene oxide (c-GO) in the cast solution. They found that the addition of c-GO improved membrane characteristics including hydrophilicity, dye rejection, and thermostability. However, the study was lack of novelty and some results are not convincing enough. Furthermore, adequate and correct mechanism analysis of the results is also lacking. Specific comments are given as follows:
1. Introduction should be further revised. For example, Line 67-70, why did the authors introduce the characteristic of the polymeric membrane again after that of the TPU membrane?
2. Page 4, lines 154-172, the introduction of water contact angle measurement should be simplified (Lines 155-166). Furthermore, the authors did not depict the measurement of the dye rejection.
3. Lines 188-190, why does the hydroxyl group peak of the GO and c-GO come from water molecules? The FTIR spectra of the GO and c-GO should be determined after drying.
4. Figure 2b, what do T, TP, TPG-1, TPG-2, TPG-3, and TPG-4 specifically refer to?
5. Figure 3, SEM images of the TPU, c-GO 1, and c-GO 2 should be provided. In addition, particle aggregation is noticeable in Figure 3b but not Figure 3c.
6. Lines 229-233, the description of XRD analysis is incorrect. For example, the broad peak (2Ó©=11-20Ëš) is only attributed to TPU and the phenomenon of peak broadening is also indiscernible in Figure 4.
7. Lines 238-246, the formation of some large holes in the c-GO 4 membrane (Figure 3c) is also responsible for the contact angle decrease. Therefore, the decrease in the contact angle, along with the increase in c-GO addition, cannot confirm the enhancement in the number of carboxyl groups on the membrane surface. Furthermore, what is the contact angle of the c-GO 0 membrane?
8. The DSC curves of the c-GO 0, c-GO 1, c-GO 2, c-GO 3, and c-GO 4 membranes should also be measured.
9. Lines 260-268, “21.81 MHL” should be changed to “21.81 LMH”. In addition, porosity of the membranes should be measured to verify the conclusion that increased porosity causes enhanced water flux.
10. With the increment of c-GO content, micrometer holes appear in the membrane. Therefore, it is not reasonable that the dye rejection enhances with the increase of c-GO content.
11. The writing of the manuscript needs improvement. Please carefully check the written errors. for example, “C-GO” should be changed to “c-GO”, in abstract; the full name of TPU is needed because this is the first time it appears, in abstract; “Polyvinyl pyrrolidone (PVP of Mw ~40000K30)” should be changed to “Polyvinyl pyrrolidone (PVP of Mw ~40000 Da)”; Line 229, “Figure 3 presents the XRD analysis” should be changed to “Figure 4 presents the XRD analysis”.
Round 2
Reviewer 1 Report
The authors have completed the required comments.
Reviewer 2 Report
The authors have correctly addressed the reviewers' concerns.